# Coupling microwave photons to a mechanical resonator using quantum interference

I.C. Rodrigues[1,2], D. Bothner[1,2] & G.A. Steele[1]*

The field of optomechanics has emerged as leading platform for achieving quantum control of macroscopic mechanical objects. Implementations of microwave optomechanics to date have coupled microwave photons to mechanical resonators using a moving capacitance. While simple and effective, the capacitive scheme suffers from limitations on the maximum achievable coupling strength. Here, we experimentally implement a fundamentally different approach: flux-mediated optomechanical coupling. In this scheme, mechanical displacements modulate the flux in a superconducting quantum interference device (SQUID) that forms the inductor of a microwave resonant circuit. We demonstrate that this flux-mediated coupling can be tuned in situ by the magnetic flux in the SQUID, enabling nanosecond flux tuning of the optomechanical coupling. Furthermore, we observe linear scaling of the single-photon coupling rate with the in-plane magnetic transduction field, a trend with the potential to overcome the limits of capacitive optomechanics, opening the door for a new generation of groundbreaking optomechanical experiments.

[1] Kavli Institute of Nanoscience, Delft University of Technology, PO Box 5046, 2600 GA Delft, The Netherlands. [2] These authors contributed equally: I. C. Rodrigues, D. Bothner *email: g.a.steele@tudelft.nl

Parametrically coupling mechanical motion to light fields confined inside a cavity has allowed for major scientific and technological breakthroughs within the recent decade[1]. Such optomechanical systems have been used for sideband-cooling of mechanical motion into the quantum ground state[2,3], for the detection of mechanical displacement with an imprecision below the standard quantum limit[4,5], for the generation of non-classical mechanical states of motion[6–8] and for the entanglement of mechanical oscillators[9,10]. As the mechanical elements can be coupled to both, light fields in the optical and in the microwave domain, current efforts using optomechanical systems target toward the implementation of a quantum link between super-conducting microwave quantum processors and optical frequency quantum communication[11,12]. Another exciting perspective of optomechanical systems is testing quantum collapse and quantum gravity models by preparing Fock and Schroedinger cat states of massive mechanical oscillators[13,14].

The state transfer fidelity between photons and phonons in optomechanical systems is determined by the coupling rate between the subsystems, and most optomechanical systems so far have single-photon coupling rates much smaller than the decay rates of the cavity. The strong-coupling regime, necessary for efficient coherent state transfer, is achieved by enhancing the total coupling rate $g = \sqrt{n_c} g_0$ through large intracavity photon numbers $n_c$[15–17]. In the optical domain, large photon numbers result in absorption that heats the mechanical mode far above the mode temperature[18]. In the microwave domain, large photon numbers result in non-equilibrium cavity noise[2,19] that is not completely understood. Both of these sources of noise limit ground state cooling and the fidelity of mechanical quantum ground state preparation. An approach to reduce these parasitic side-effects is to increase the single-photon coupling rate $g_0$ significantly. Doing so, optomechanics could even reach the single-photon strong-coupling regime, where the optomechanical system acquires sufficient nonlinearity from the parametric coupling such that non-Gaussian mechanical states can be directly prepared by coherently driving the system[20,21].

In the microwave domain, the most common approach to build an optomechanical system is to combine a superconducting microwave LC circuit with a metallized suspended membrane or nanobeam as mechanical oscillator. The devices are constructed in a way that the displacement of the mechanical oscillator changes the capacitance of the circuit $C(x)$ and hence its resonance frequency $\omega_0(x) = 1/\sqrt{LC(x)}$. In this configuration, however, the single-photon coupling rate is limited to $g_0 \leq \frac{\omega_0}{2} \frac{x_{zpf}}{d}$ with the zero-point fluctuation amplitude $x_{zpf}$ and the capacitor gap $d$. Current devices are highly optimized, but still achieve typically only $x_{zpf}/d \approx 10^{-7}$ for a parallel plate capacitor gap of $d = 50$ nm and it is extremely challenging to increase $g_0$ beyond 300 Hz with this approach.

Here, we realize a fundamentally different approach for a microwave optomechanical device by incorporating a suspended mechanical beam into the loop of a superconducting quantum interference device (SQUID). The SQUID itself is part of a superconducting LC circuit and essentially acts as an inductor, whose inductance depends on the magnetic flux threading through the loop. In this article, we demonstrate that this magnetic flux-mediated inductive coupling scheme provides quickly tunable single-photon coupling rates[22,23], which in addition scale linearly with a magnetic field applied in the plane of the SQUID loop[24]. In contrast to the capacitive approach, the coupling rates in flux-mediated optomechanics are not limited by geometric and technological restrictions and there is a realistic prospective for achieving the optomechanical single-photon strong-coupling regime.

## Results

**Concept and device.** The concept of coupling mechanical resonators to SQUIDs has been developed in many works[25–28], including earlier experimental work with DC SQUIDs[29,30]. Recently, this concept was extended theoretically to optomechanics[24], describing a way using SQUIDs to achieve strong and tunable optomechanical coupling between a vibrating beam and a superconducting cavity. The circuit used here for its realization is schematically shown in Fig. 1a. The idea is based on transducing mechanical displacement to magnetic flux, which in turn modulates the effective inductance of a SQUID and therefore the resonance frequency of the LC circuit hosting it. To achieve this transduction from displacement to flux, a part of the SQUID loop is suspended and the device is exposed to an external magnetic field $B_\parallel$ applied parallel to the device plane. The suspended loop part acts as a mechanical beam resonator and its vibrational motion, perpendicular to the device plane, will create an effective SQUID area perpendicular to the applied field $B_\parallel$, i.e., couple a net magnetic flux into the loop.

The inductance $L(\Phi_b)$ of an LC circuit containing a SQUID depends on the magnetic flux threading the SQUID loop, and translates to a flux-dependent resonance frequency

$$\omega_0(\Phi_b) = \frac{1}{\sqrt{L(\Phi_b)C}}. \tag{1}$$

When the displacement of a mechanical oscillator is transduced to additional flux, an optomechanical interaction between mechanical mode and cavity resonance frequency emerges and the single-photon coupling rate is given by ref. [24]

$$g_0 = \frac{\partial \omega_0}{\partial \Phi} \Phi_{zpf} = \frac{\partial \omega_0}{\partial \Phi} \gamma B_\parallel l x_{zpf}. \tag{2}$$

The first term $\partial \omega_0 / \partial \Phi$ corresponds to the responsivity of the SQUID cavity resonance frequency to small changes of flux through the loop and allows for very fast tuning of $g_0$. The second term $\Phi_{zpf} = \gamma B_\parallel l x_{zpf}$ is the magnetic flux fluctuation induced in the SQUID by the mechanical zero-point fluctuations $x_{zpf}$ of the beam with length $l$ and scales linearly with an in-plane magnetic field $B_\parallel$, cf. Fig. 1. The scaling factor $\gamma$ accounts for the mode shape of the mechanical oscillations and is on the order of 1.

The microwave SQUID cavity in our experiment is made of a single 20-nm thick layer of sputtered aluminum on a silicon substrate and it contains a SQUID consisting of two constriction type Josephson junctions placed in parallel in a $21 \times 5$ μm$^2$ closed loop. An optical image of the device is shown in Fig. 1b and an electron microscope image of the SQUID loop in Fig. 1c, the fabrication is detailed in the Supplementary Note 1. The capacitance of the LC circuit is formed by two interdigitated capacitors $C$ to ground and a coupling capacitor $C_c$ to the center conductor of a coplanar waveguide feedline. In addition, to the SQUID inductance $L_S = L_J/2$ with the Josephson inductance $L_J$ of a single junction, there are two linear inductances $L$ built into the circuit in order to dilute the nonlinearity of the cavity, arising from the non-linear Josephson inductance. By this measure we achieve an anharmonicity of approximately 15 Hz per photon and enable the multi-photon coupling rate enhancement $g = \sqrt{n_c} g_0$ of linearized optomechanics.

The cavity is side-coupled to a coplanar waveguide microwave feedline, which is used to drive and read-out the cavity response by means of the transmission parameter $S_{21}$. The device is mounted into a radiation tight metal housing and attached to the mK plate of a dilution refrigerator with a base temperature of approximately $T_b = 15$ mK, cf. Supplementary Note 2. Without any flux biasing, the cavity has a resonance frequency $\omega_0 = 2\pi \times 5.221$ GHz and a

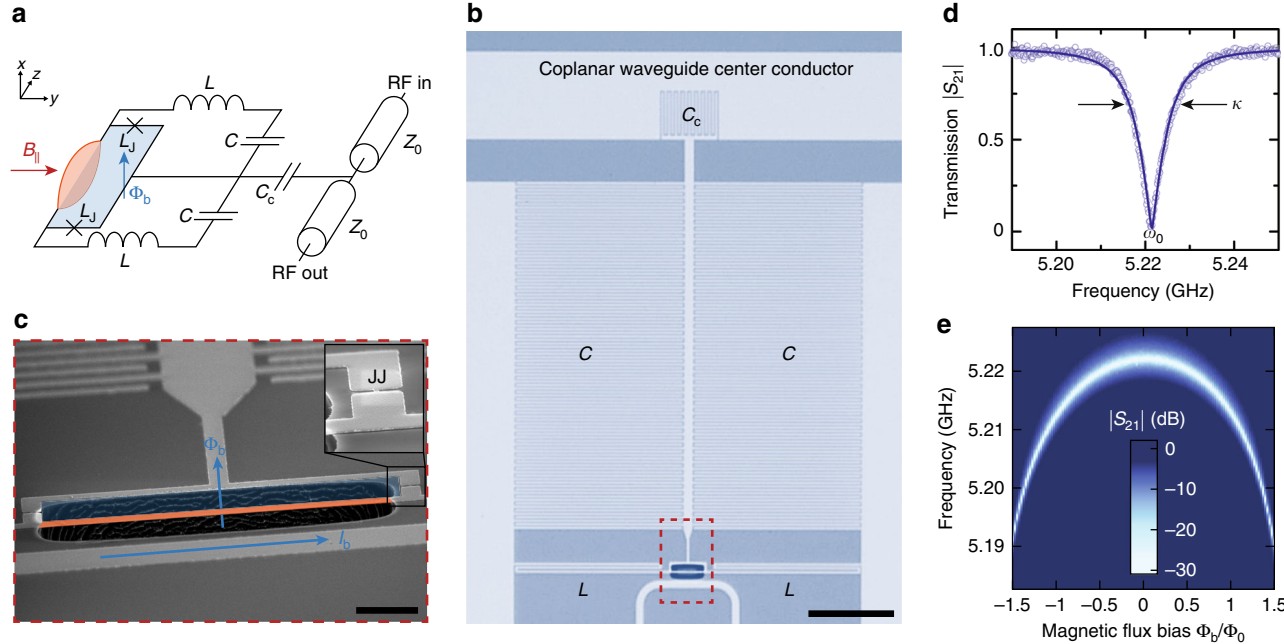

**Fig. 1** A superconducting microwave circuit with magnetic-flux-mediated optomechanical coupling to a mechanical oscillator. **a** Circuit schematic of the device. The LC circuit is capacitively coupled to a microwave transmission line with characteristic impedance $Z_0$ by means of a coupling capacitor $C_c$. In addition to the linear capacitors $C$ and inductors $L$, a superconducting quantum interference device (SQUID) is built into the circuit, consisting of two Josephson junctions with inductance $L_J$ in a closed superconducting loop, of which a part is suspended and free to move perpendicular to the circuit plane. To bias the SQUID with magnetic flux $\Phi_b$, a magnetic field can be applied perpendicular to the circuit plane. Motion of the mechanical element is transduced into modulations of the bias flux by a magnetic in-plane field $B_{||}$. An optical micrograph of the circuit is shown in (**b**), light gray parts correspond to a 20nm thick layer of aluminum, dark parts to silicon substrate. The black scale bar corresponds to 50 μm. The red dashed box shows the region, which is depicted in a tilted scanning electron micrograph in (**c**), showing the SQUID loop with the released aluminum beam. The bias flux through the SQUID loop $\Phi_b$ can be changed by a bias current $I_b$ sent through the on-chip flux bias line. The black scale bar corresponds to 3 μm. The inset shows a zoom into one of the constriction type Josephson junctions (JJs). In (**d**) the cavity resonance is shown, measured by sending a microwave tone to the microwave feedline and detecting the transmitted signal $S_{21}$. A fit to the data points (circles), shown as line, reveals a resonance frequency of $\omega_0 = 2\pi \times 5.221$ GHz and a linewidth $\kappa = 2\pi \times 10.5$ MHz. Panel (**e**) shows color-coded the tuning of the cavity resonance absorption dip with magnetic bias flux in units of flux quanta $\Phi_b/\Phi_0$, measured at $B_{||} = 1$ mT. Due to a large loop inductance of the SQUID, the arch exceeds a single flux quantum, for details see Supplementary Note 3

linewidth $\kappa = 2\pi \times 9$ MHz, which at the same time corresponds to the external linewidth $\kappa \approx \kappa_e$ due to being deep in the over-coupled regime, cf. the cavity resonance curve shown in Fig. 1d. When magnetic flux is applied to the SQUID loop by sending a current to the chip via the on-chip flux bias line, the cavity resonance frequency is shifted toward lower values due to an increase of the Josephson inductances inside the SQUID. The flux-dependent transmission $|S_{21}|(\Phi)$ is shown in Fig. 1e and a total tuning of about 30 MHz can be achieved, mainly limited by a non-negligible SQUID loop inductance of the SQUID and the dilution of the Josephson inductance by $L_J/(L + L_J) \approx 0.01$, see also Supplementary Note 3. The largest flux responsivities we could achieve here were approximately $\partial\omega_0/\partial\Phi = 2\pi \times 70$ MHz/$\Phi_0$.

We note, that the cavity linewidth $\kappa$ depends on the flux bias and both, the linewidth and the shape of the resonance frequency flux tuning depend slightly on the magnetic in-plane field. Also, the observation that the SQUID cavity tuning curve shown in Fig. 1e extends beyond $\pm\Phi_0/2$ might be surprising at first, but is explained by a non-negligible SQUID loop inductance relative to the Josephson inductance. A detailed discussion with additional data on both these effects is given in the Supplementary Note 3.

The mechanical oscillator is a $20 \times 1\,\mu m^2$ large aluminum beam and is suspended as result of releasing part of the superconducting loop forming the SQUID by removing the underlying silicon substrate in an isotropic reactive ion etching process[31]. The beam has a total mass $m = 1$ pg and its fundamental out-of-plane mode

oscillates at a frequency $\Omega_m = 2\pi \times 7.129$ MHz with an intrinsic mechanical damping rate of $\Gamma_m \approx 2\pi \times 8$ Hz or quality factor $Q_m = \Omega_m/\Gamma_m \approx 9 \times 10^5$, which is exceptionally high for a mechanical oscillator made from a sputter-deposited metal film. From the mass and resonance frequency, the zero-point motion of the oscillator is estimated to be $x_{zpf} = \sqrt{\frac{\hbar}{2m\Omega_m}} = 33$ fm.

**Detection of mechanical displacement with a SQUID cavity.** The mechanical beam can be coherently driven by Lorentz-force actuation using the on-chip flux bias line. When a current is sent through the bias line, magnetic flux is coupled into the SQUID loop and a circulating loop current is flowing through the mechanical oscillator. We apply a current $I_\Omega(t) = I_{dc} + I_0 \cos\Omega t$ with $\Omega \approx \Omega_m$, where the DC component $I_{dc}$ is simultaneously biasing the SQUID and—in presence of an in-plane magnetic field $B_{||}$—exerting a constant Lorentz-force to the beam. The oscillating part $I_0 \cos\Omega t$ modulates the total Lorentz-force $F_L(t) = F_{dc} + F_0 \cos\Omega t$ around the equilibrium value $F_{dc}$ and effectively drives the mechanical oscillator. The concept is illustrated in Fig. 2a, for more details, cf. Supplementary Note 4.

The resulting mechanical motion modulates the cavity resonance frequency and generates sidebands at $\omega_d \pm \Omega$ to a microwave signal sent into the cavity at $\omega_d = \omega_0$, cf. the schematic in Fig. 2b. By sweeping $\Omega$ through $\Omega_m$ and down-converting the sidebands generated at $\omega = \omega_0 - \Omega$ and $\omega = \omega_0 + \Omega$, we measure the

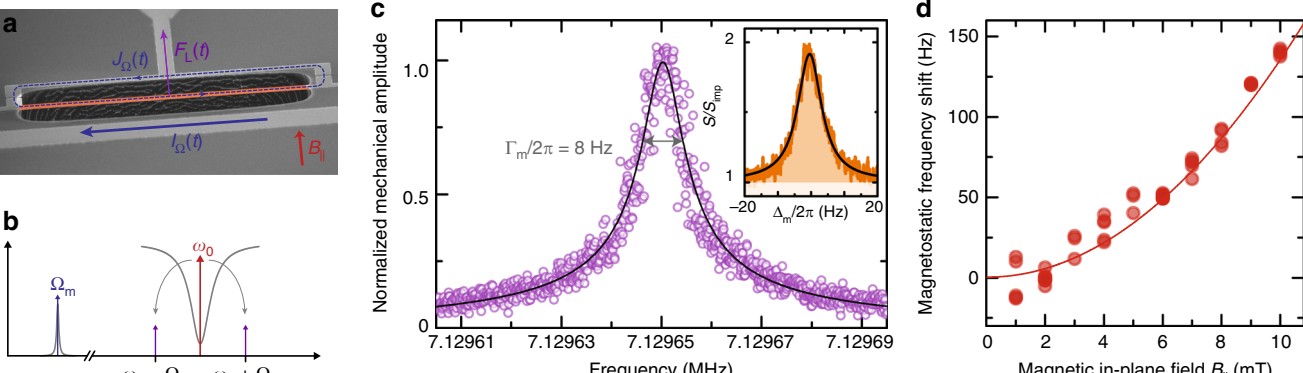

**Fig. 2** Detection of mechanical motion using a superconducting SQUID cavity interferometer and observation of magnetostatic spring stiffening. **a** Schematic of coherently driving the mechanical oscillator by means of the Lorentz-force. The current sent through the bias line has a DC component to bias the SQUID with a flux $\Phi_b$. This generates a circulating current $J$ in the SQUID loop. In addition, an oscillating current is sent through the line with a frequency close to the mechanical oscillator resonance frequency $\Omega \approx \Omega_m$. Thus, the loop current through the mechanical beam oscillates correspondingly, leading to an oscillating Lorentz-force $F_L(t)$ due to the presence of the magnetic in-plane field $B_{||}$. The mechanical motion modulates the total magnetic flux through the SQUID loop and hence the cavity resonance frequency. When a resonant coherent microwave tone is sent into the cavity, the mechanical oscillations generate sidebands at $\omega = \omega_0 \pm \Omega_m$, cf. panel (**b**), which are observed to detect the mechanical motion. In (**c**), the down-converted sideband signal is shown during a sweep of the excitation frequency $\Omega$. Circles are data, the line is a Lorentzian fit and both are normalized to the maximum of the fit curve. The inset depicts the down-converted sideband thermal noise spectral density in absence of a coherent driving force, normalized to the background noise floor. Orange line are data, black line is a Lorentzian fit. The contribution from the background noise is shaded in white and the contribution from the mechanical displacement noise in orange. The experimental settings for these measurements were $B_{||} = 9$ mT and $\partial\omega_0/\partial\Phi \sim 2\pi \times 20$ MHz/$\Phi_0$. When increasing the magnetic in-plane field, we observe a shift of the mechanical oscillator resonance frequency, shown in panel (**d**). This frequency shift is induced by a position-dependent contribution to the Lorentz-force and corresponds to a magnetostatic stiffening of the mechanical spring constant. The circles are data and the line corresponds to a theoretical curve with $\delta\Omega_m \propto B_{||}^2$

mechanical resonance as shown in Fig. 2c. This interferometric detection scheme of displacement can also be used to detect the thermal motion of the mechanical oscillator. At the dilution refrigerator base temperature $T_b = 15$ mK, we expect a thermal mode occupation of the beam of approximately $n_{th} = k_B T_b/(\hbar\Omega_m) \approx 46$ phonons with $k_B$ being the Boltzmann constant. In the inset of Fig. 2c we show the down-converted sideband power spectral density $S$ of the cavity output field, normalized to the background noise, without any external drive applied to the mechanical oscillator. On top of the imprecision noise background $S_{imp}$ of the measurement chain, a Lorentzian peak with a linewidth of $\sim 8$ Hz is visible, generated by the residual thermal motion of the beam.

When we sweep the magnetic in-plane field $B_{||}$, we observe an increase of the mechanical resonance frequency as shown in Fig. 2d induced by Lorentz-force backaction[30]. Complementary to the electrostatic spring softening in mechanical capacitors with a bias voltage, this effect can be understood as a magnetostatic spring stiffening. When the mechanical oscillator is displaced from its equilibrium position, an additional magnetic flux is coupled into the SQUID loop, which leads to an adjustment of the circulating current $J$ to fulfill fluxoid quantization inside the loop. Hence, the Lorentz-force $F_L \propto B_{||}J$ will change accordingly and therefore has a contribution dependent on the mechanical position. For small mechanical amplitudes and circulating currents not too close to the critical current of the Josephson junctions, this position dependence will be linear, causing a frequency shift $\delta\Omega_m \propto B_{||}^2$, cf. the discussion in the Supplementary Note 4.

**Tuning $g_0$ with the SQUID flux operation point**. When a magnetic bias flux is applied to the SQUID, not only the cavity resonance frequency changes, but also the flux responsivity $\partial\omega_0/\partial\Phi$. As the optomechanical single-photon coupling rate is directly proportional to the responsivity, it can in principle

be switched on and off on extremely short timescales or can be dynamically controlled by flux modulating the SQUID. We demonstrate this tuning of the single-photon coupling rate with bias flux by determining $g_0$ for different values of $\Phi_b/\Phi_0$.

One possibility to determine the multi-photon coupling rate $g$ in an optomechanical system is to perform the experimental scheme of optomechanically induced transparency[32,33]. For this scheme, a strong coherent microwave tone is driving the cavity on the red sideband $\omega_d = \omega_0 - \Omega_m$ and a weak probe tone is sent to the cavity around $\omega_p \approx \omega_0$. The two tones interfere inside the cavity, resulting in an amplitude beating with the frequency difference $\Omega = \omega_p - \omega_d$. If the beating frequency is resonant with the mechanical mode, the radiation pressure force resonantly drives mechanical motion which, in turn, modulates the cavity resonance and the red sideband drive tone. The modulation generates a sideband to the drive at $\omega = \omega_d + \Omega$, which interferes with the original probe field in the cavity. This interference effect opens up a narrow transparency window within the cavity response, which has the shape of the mechanical resonance, modified by the dynamical backaction of the red sideband tone. For $\omega_d = \omega_0 - \Omega_m$ the magnitude of the transparency window $|S_m|$ with respect to the depth of the cavity resonance dip $|S_c|$ is directly related to the coupling rate via

$$\frac{|S_m|}{|S_c|} = \frac{4g^2}{\kappa\Gamma_{eff}} \qquad (3)$$

where $\Gamma_{eff} = \Gamma_m + \Gamma_o$ is the width of the transparency window, given by the intrinsic mechanical damping $\Gamma_m$ and the optomechanically induced damping $\Gamma_o$. In combination with a careful calibration of the intracavity photon numbers $n_c$, we use this approach to get an estimate for the single-photon coupling rate $g_0 = g/\sqrt{n_c}$. More details on the photon number calibration and the extraction of $g$ from the OMIT data are given in the Supplementary Note 5.

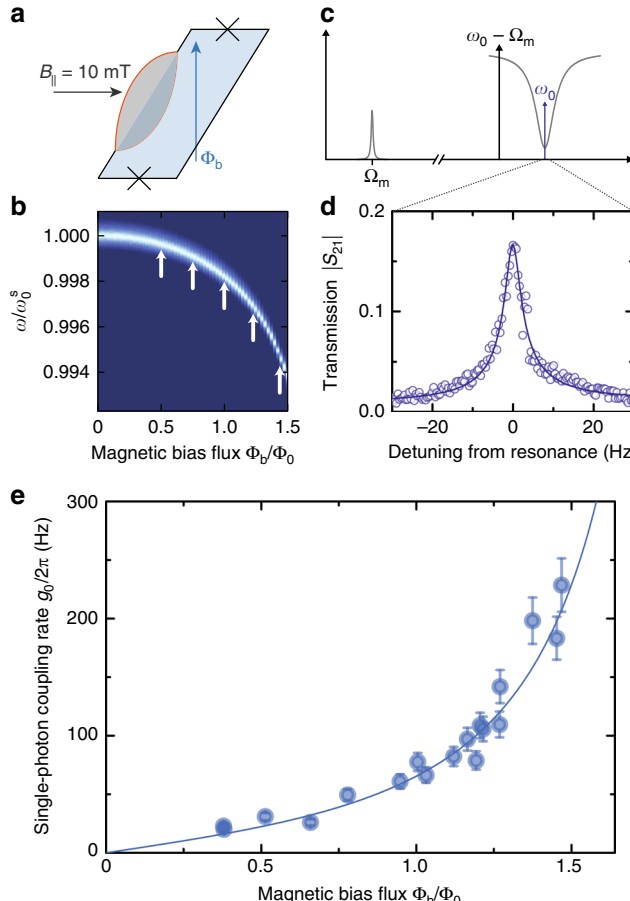

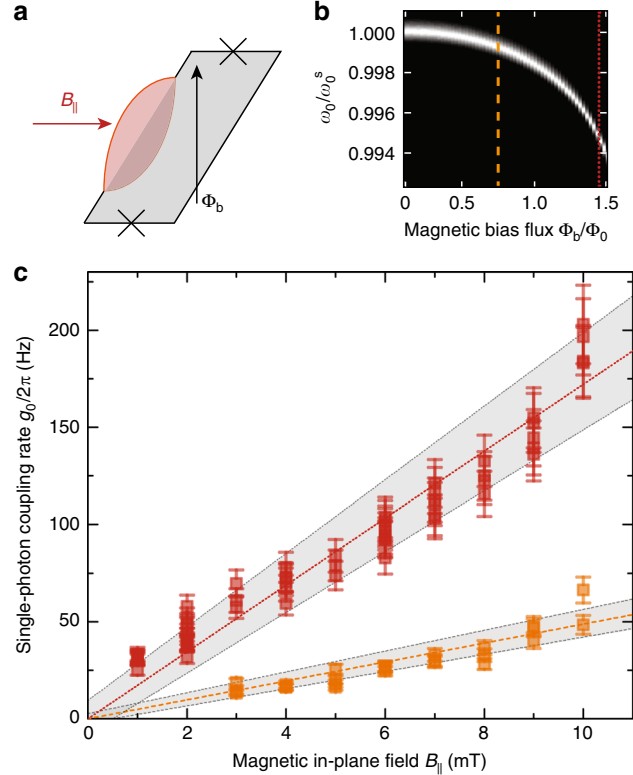

**Fig. 4** Scaling up the optomechanical single-photon coupling rate with the applied in-plane magnetic field. **a** Representation of the applied magnetic field components to the SQUID loop. During the experiment, the cavity flux responsivity was fixed at two different values by adjusting the flux bias point $\Phi_b$. In addition to this constant parameter, the in-plane magnetic field $B_{\parallel}$ was swept from 1 to 10 mT in steps of 1 mT. The transmission $|S_{21}|$ depending on the normalized bias flux is shown in (**b**) for $B_{\parallel} = 1$ mT (black: 0 dB, white: −30 dB). The two different set-points represented as orange dashed and red dotted lines, respectively, correspond to a flux responsivity of $\sim 17$ MHz/$\Phi_0$ and $\sim 60$ MHz/$\Phi_0$. Posterior to tuning the cavity to the desired working point, an OMIT experiment was performed and the single-photon coupling rate of the system was extracted. The experimental procedure was repeated in increasing steps of 1 mT of in-plane field. The resulting single-photon coupling rates $g_0$ are shown in (**c**) as squares. The dashed and dotted lines show theoretical lines and the gray areas consider uncertainties in the flux responsivity of 10% and a possible in-plane field offset of ±0.5 mT. Error bars consider 10% uncertainty in the extracted values, cf. Supplementary Note 6

**Fig. 3** Tuning the optomechanical single-photon coupling rate by changing the flux operating point of the SQUID. **a** Schematic of the applied magnetic field components to the SQUID loop. The in-plane magnetic field $B_{\parallel}$ is set by means of a cylindrical coil wrapped around the whole sample mounting. During this experiment, it was kept constant at $B_{\parallel} = 10$ mT. In addition, an out-of-plane magnetic field was varied by changing the current sent through the on-chip flux bias line, generating a magnetic bias flux $\Phi_b$. **b** As consequence of changing the amount of flux threading the SQUID loop, both the resonance frequency as well as the flux responsivity $\partial\omega_0/\partial\Phi$ of the cavity are changed. The plot shows $|S_{21}|$ ($B_{\parallel} = 1$ mT), the frequency axis is normalized to the sweetspot resonance frequency $\omega_0^s$, the color code is given in Fig. 1e. The white arrows represent the points, for which we performed the measurement scheme of optomechanically induced transparency (OMIT) as shown schematically in (**c**). A coherent drive tone is set to the red sideband of the SQUID cavity ($\omega_d = \omega_0 - \Omega_m$), while a small probe tone is scanning the cavity resonance $\omega_p \approx \omega_0$. As result of an interference effect, a transparency window in the transmitted signal $S_{21}$ is visible around $\omega_d + \Omega_m$, as shown in (**d**), where the circles represent the data and the line the corresponding fit curve. By setting the cavity to different flux bias points (white arrows in (**b**)), we change the cavity flux responsivity and therefore the single-photon optomechanical coupling rate $g_0 \propto \partial\omega_0/\partial\Phi$. From the magnitude of the transparency window, $g_0$ can be extracted for each flux bias point. The result is plotted in (**e**) as circles. The line is the theoretical curve as described in the main text. Error bars in **e** show the estimated uncertainty of 10%, cf. Supplementary Note 6

When performing this experiment for several different flux bias points, we find a clear increase of $g_0$ with the cavity flux responsivity. The experimental scheme and the obtained single-photon coupling rates for a constant in-plane field of $B_{\parallel} = 10$ mT are shown in Fig. 3. In Fig. 3e we also plot as line the theoretical curve, where the only free parameter is the scaling factor $\gamma = 0.9$,

taking into account the mode shape of the mechanical oscillations. All other contributions to the calculations were obtained from independent measurements, such as the bias flux dependence of the cavity frequency, the mechanical resonance frequency and estimations for the beam length and its mass. The largest single-photon coupling rate we achieve here $g_0 \approx 2\pi \times 230$ Hz is comparable to the best values obtained for highly optimized capacitively coupled devices. As it is possible to achieve responsivities of several GHz/$\Phi_0$ with SQUID cavities[34,35], we expect that with an optimized cavity it is possible to boost the single-photon coupling rates to several kHz per mT of in-plane field. This optimization with respect to $\partial\omega_0/\partial\Phi$ can be achieved by reducing either the SQUID loop inductance or the Josephson junction critical current or by a combination of both.

From the Kerr-nonlinearity of our device $\chi/2\pi \sim 120$ Hz for the largest measured responsivity, we estimate intracavity photon numbers up to $\sim 10^5$ to be compatible with the cavity, which corresponds to maximally achievable multi-photon coupling rates

of $g = 2\pi \times 70\,\mathrm{kHz}$ and cooperativities of $\mathcal{C} = \frac{4g^2}{\kappa \Gamma_m} \sim 300$. Due to the large loop inductance of the used SQUID, however, the cavity is operated in a metastable flux branch (see Supplementary Note 3) and we were limited to work with $n_c \sim 150$ intracavity photons at the largest flux responsivities before switching to the stable flux branch, which limited $g$ and $\mathcal{C}$ to $g \sim 2\pi \times 3\,\mathrm{kHz}$ and $\mathcal{C} \sim 0.5$ in current experiments.

**Linear scaling of $g_0$ with the in-plane transduction field**. As an ultimate experimental signature that our device transduces mechanical displacement to magnetic flux, we investigate the scaling of the optomechanical coupling rate with magnetic in-plane field $B_\parallel$. Therefore, we performed the scheme of optomechanically induced transparency for constant values of flux responsivity $\partial\omega_0/\partial\Phi$ but for varying in-plane magnetic field. First, we chose a fixed responsivity of about $\partial\omega_0/\partial\Phi \approx 2\pi \times 17\,\mathrm{MHz}/\Phi_0$ and then adjusted $B_\parallel$ in steps of 1 mT. For each $B_\parallel$ we perform several OMIT experiments and extract the single-photon coupling rates as described above. This whole scheme was repeated for $\partial\omega_0/\partial\Phi \approx 2\pi \times 60\,\mathrm{MHz}/\Phi_0$.

The resulting single-photon coupling rates are shown in Fig. 4 and follow approximately a linear increase with in-plane magnetic field. The theoretical lines correspond to independent calculations based on the flux dependence of the cavity, and the parameters of the mechanical oscillator. The data clearly demonstrates that we observe a flux-mediated optomechanical coupling, a system in which the coupling rates can be further increased with higher magnetic in-plane fields. In the current setup, we were limited to the field range up to 10 mT. Due to an imperfect alignment between the chip and the in-plane field, a considerable out-of-plane component was present and, most probably by introducing vortices, strongly influenced the properties of the cavities above $B_\parallel = 10\,\mathrm{mT}$. Using a vector magnet to compensate for possible misalignments will allow to go up to about 100 mT with thin film aluminum devices[36,37] resulting in coupling rates of several hundreds of kHz. When extending the used material to other superconductors such as Niobium or Niobium alloys such as NbTiN, where similar constriction type SQUIDs have recently been used for tunable resonators[38], the possible field range for the in-plane field increases up to the Tesla regime[39]. We believe that the maximum applicable in-plane field is the most relevant practical limitation for the scaling of the optomechanical coupling rate but it is unknown at this point how large it can be made while preserving high quality factor SQUID cavities and mechanical beams using other superconductors.

## Discussion

With the realization of flux-mediated optomechanical coupling reported in this article, the door is open for a new generation of microwave optomechanical systems. The single-photon coupling rates achieved with this first device are already competing with the best electromechanical systems and can be boosted toward the MHz regime by optimizing the flux responsivity and applying higher magnetic in-plane fields. In addition, reducing the cavity linewidth to values of $\leq 100\,\mathrm{kHz}$ will lead us into the single-photon strong-coupling regime, where a new type of devices and experiments can be realized, amongst others the realization of a new class of microwave qubits, where the nonlinearity arises from the coupling to a mechanical element, the generation of mechanical quantum states or optomechanically induced photon blockade[21]. The coupling mechanism between a mechanical oscillator and a microwave circuit, which we realized here, has also been intensely discussed in the context of superconducting

flux and transmon qubits instead of linear cavities[25,28,40] and could now be implemented using circuits with a large Josephson nonlinearity leading to a new regime of quantum control of macroscopic mechanical objects. Utilizing quantum states of mechanical resonators as resource for quantum information and quantum sensing technologies is a promising approach due to the typically very long lifetimes of mechanical excitations and the possibility to couple mechanical systems simultaneously to microwave cavities and optical fields[12,41,42] and at the same time can be used to test quantum collapse models and decoherence mechanisms in the presence of large masses, i.e., test quantum mechanics itself with massive quantum states[13,14,43].

## Data availability

All raw data and processed data as well as supporting code for processing and figure generation is available in Zenodo with the identifier https://doi.org/10.5281/zenodo.3470882.

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

## Acknowledgements

The authors thank R. Norte for help with the device fabrication and M.D. Jenkins for support with the data acquisition software. This research was supported by the Netherlands Organisation for Scientific Research (NWO) in the Innovational Research Incentives Scheme – VIDI, project 680-47-526, the European Research Council (ERC) under the European Union's Horizon 2020 research and innovation program (grant agreement No. 681476 - QOMD) and from the European Union's Horizon 2020 research and innovation program under grant agreement No. 732894 - HOT.

## Author contributions

I.C.R. and D.B. designed and fabricated the device, performed the measurements, and analyzed the data. G.A.S. conceived the experiment and supervised the project. All authors wrote the paper and discussed the results.

## Competing interests

The authors declare no competing interests.
