## [Peer Review File · Nature Communications]

Reviewers' Comments:

Reviewer #1:

Remarks to the Author:

Dear Editor,

the work by the Steele group describes the realization of an opto-mechanic interaction using a superconducting microwave circuit. In contrast to earlier realizations, the authors use an approach based on a displacement sensitive inductance instead of the established displacement dependent capacitance approach. The authors hereby demonstrate a bare optomechanical coupling of about 260 Hz which is on-par with highly optimized structures based on a capacitive implementation of the coupling. This result is new, timely and important and shows that using this concept the ultra-strong coupling regime is within reach. The manuscript is well written, a pleasure to read, and the SI gives all necessary details.

I will strongly support the publication of the manuscript after the authors addressed the following points:

- 1) In Figure 1 the authors discuss the performance of their microwave resonator. I suppose that these parameters will depend on the applied magnetic field. The authors should include (e.g. in their SI) more data, in particular data corresponding to the highest coupling rate $g_0/2\pi = 250$ Hz. It would be extremely beneficial to document the performance of the microwave resonator under these conditions in order to judge the possibility to go to higher fields.
- 2) Figure 1e) the authors should include the discussion about the large β_L in the main text. It is initially very surprising to find a "flux-periodicity" of $3\Phi_0$ in the panel. As this panel is the key information to compute the "responsivity" of the microwave resonator, the authors should also include the actual bias current /bias field in the SI. One more question in this regard, if the data shown is taken around $B_{\text{parallel}} = 0$, then the authors should give a statement if the same responsivity is found for $B_{\text{parallel}} = 10\text{mT}$.
- 3) Also related to point 2, the authors discuss various calibration techniques for Φ_b in the SI. As Φ_b seems to be generated by the on-chip flux bias line, I encourage the authors to numerically compute the magnetic field (e.g. using Biot-Savard) from the bias current. Does this way of calibration fits with the γ_L stated in the SI ? This represents a very critical point as the responsivity is directly connected to the magnetic field / flux calibration.
- 4) Page 2 paragraph starting with "The concept of coupling mechanical ...". Theoretical proposals of squid induced optomechanical coupling date back to 2007 (Phys Rev B 76, 014511 (2007), Phys Rev B 78, 104516 (2008). Phys Rev A 93, 022510 (2016)) . Those works should also get credit.
- 5) Thermal noise spectrum given in Fig 2c (inset). Given the excitation scheme with the flux bias line, the authors should have a calibration technique to relate the magnetic flux fluctuations to the frequency fluctuations of the microwave resonator. To be specific, as the authors know the resonance frequency modulation for a given VNA "current amplitude", they should be able to convert this amplitude to a flux amplitude (again using e.g. modeling of the stray-field of the line). This should give directly the calibration factor relevant for converting the flux fluctuations to a g_0 . What value of g_0 would the authors get for this spectrum and does it fit to the electromechanically induced transparency method?
- 6) In the context of point 5. Do the authors have any indications, that their mechanical resonator is in thermal equilibrium with the temperature of the fridge? Please comment.
- 7) When the authors discuss figure 2, they should state the flux-bias conditions responsivity and B_{parallel} . B_{parallel} of course only for panel c .
- 8) SI: Chapter S2.A: I noticed 2 circulators, a coupler and a filter in the line to the cryogenic hemt amplifier. In what sense are those included in the attenuation discussion? Typically, those can account (depending on model) for losses in the range of 2dB. How does that affect g_0 ?
- 9) Chapter S5 B: given the parameters stated in the manuscript for the $g_0/2\pi = 250\text{Hz}$ case, the "optical" damping described in (S49) should be on the order of the linewidth. Do the authors find

the expected signatures of opto-mechanical “cooling”? This should also be directly visible in the transparency data. I suggest to extract Γ_{eff} and plot it vs the photon number to confirm the expectations.

10) Chapter S5 C: To the best of my knowledge, the Supplemental Information of the paper “Weiss et al., Science 330, 1520 (2010)” and the SI of “Zhou, X. et al., Nat Phys 9, 179 (2013)” contain a full description of the interference effect including both sidebands. As far as I understand, the authors in Weiss et al. are not using the resolved assumption in S23. Same is should be the case for S27 in Zhou. I wonder how the proposed fit methods in the manuscript by Rodrigues et al. relate to this transmission model. Please comment.

11) Chapter S5 D 1: Just out of curiosity, why do the authors stop the pulse tube ? They should spend a sentence what they prevent here experimentally.

12) Fig S9. The authors should include the squid bias conditions, the photon number and the B_{parallel} for this dataset. Is the extracted effective linewidth consistent with the Γ_{eff} expected by (S49).

Technical points:

13) Fig S9: typo at “b Dta a and fit...” should read “Data...”

14) SI Page 8: second last line. There is a 2π missing in the equation of ω_2

Reviewer #2:

Remarks to the Author:

This submission by Rodrigues et al. reports on a new scheme for coupling microwave photons to mechanical resonators with prospects for overcoming limitations to coupling rates inherent to the conventional technique. Specifically, the authors use a device in which the mechanical resonator's displacement modulates the flux through a SQUID integrated within a resonant microwave circuit. This scheme allows for the tuning of the flux-mediated coupling, with single photon rates reaching values comparable to the state of the art.

The experimental work is of high quality and is clearly presented both in the text and in a series of nice figures. The analysis of the data is done with care and appears valid. The resulting conclusions are well-supported and based on valid assumptions. A thorough supplementary section describes the experimental details. In summary, the paper provides an excellent proof of principle for a flux-mediated optomechanical device. Although the experiment presents some limitations (e.g. the availability of only 10 mT of applied in-plane field, putting an experimental ceiling on the achievable optomechanical coupling), the authors attempt to show the promise of the technique and possibilities that it may open.

For all of these reasons, I find the paper appropriate for publication once the authors have addressed a few key questions, mostly related to the proof-of-principle nature of the result, i.e. the authors should better spell out what the ultimate limitations of this technique are and what exactly could be achieved with the ideal device.

1) In the first paragraph of page 2, the authors state that unlike capacitive optomechanical cavities, which are limited by geometric and technological restrictions, their cavities may be able to reach the single-photon strong coupling regime. What are then the restrictions that limit their flux-mediated scheme? How far, in principle, could such a scheme push the coupling rates? What would have to be optimized?

2) As stated in the 2nd to last paragraph of page 5, the authors largest achieved single-photon coupling rate is $2\pi \cdot 230$ Hz, which is comparable to the rates achieved by best conventional devices. They then claim that it should be possible to boost this rate by about 3 orders of magnitude, because SQUID cavities can be made with responsivities of several GHz/ Φ_0 . From

Figure 1e, it looks like their cavity has about $10 \text{ MHz}/\Phi_0$. Given that a 3 order of magnitude boost in coupling is a big claim, it should be further justified. What steps would have to be taken to achieve such the required responsivity? What has to be improved relative to the presented device? Can this be realized in a device with the functionality of the one demonstrated in the manuscript? What are the limitations?

3) In the conclusion, the authors restate the possibility of using their scheme to achieve single-photon strong coupling. However, the description of what this means and what could be achieved with such devices is rather vague and devoid of references. The non-expert reader would benefit from a bit more clarification. What exactly are the "new type of decices and experiments" that can be realized? Why are microwave qubits with a nonlinearity induced by mechanics useful/interesting? What is the point of generating mechanical quantum states or photon blockade?

Reply to the reviewer comments and suggestions

Reply to reviewer #1:

Reviewer #1:

"This result is new, timely and important and shows that using this concept the ultra-strong coupling regime is within reach. The manuscript is well written, a pleasure to read, and the SI gives all necessary details."

Reply:

We thank the reviewer for their careful reading of our manuscript and their positive assessment and appreciation of our work.

Reviewer #1:

"1) In Figure 1 the authors discuss the performance of their microwave resonator. I suppose that these parameters will depend on the applied magnetic field. The authors should include (e.g. in their SI) more data, in particular data corresponding to the highest coupling rate $g_0/2\pi = 250$ Hz. It would be extremely beneficial to document the performance of the microwave resonator under these conditions in order to judge the possibility to go to higher fields."

Reply:

Yes indeed, in our experiment the cavity linewidth is dependent on magnetic field. In our device, however, the linewidth is predominantly given by the external linewidth and a reliable extraction of the internal linewidth is essentially impossible due to the large mismatch between κ_e and κ_i in combination with cable resonances. The apparent external linewidth on the other hand is not directly dependent on the magnetic field, but it is mainly influenced by the position of the cavity resonance within the cable resonances of our setup.

And although a quantitative discussion of this effect is very difficult, we included a new figure, showing the linewidth dependence on the two magnetic fields, and a corresponding discussion into the new version of the manuscript SI.

Regarding the possibility to go to higher fields, it will be crucial to align the in-plane field perfectly with the chip, as for now we attribute the main harm arising from the in-plane field to its out-of-plane component, limiting us to 10 mT. For now, however, we cannot really separate the effects of the two field components in our experiment.

Reviewer #1:

"2) Figure 1e) the authors should include the discussion about the large β_L in the main text. It is initially very surprising to find a "flux-periodicity" of $3\Phi_0$ in the panel. As this panel is the key information to compute the "responsivity" of the microwave resonator, the authors should also include the actual bias current /bias field in the SI. One more question in this regard, if the data shown is taken around $B_{\text{parallel}} = 0$, then the authors should give a statement if the same responsivity is found for $B_{\text{parallel}} = 10\text{mT}$."

Reply:

The data for the flux dependence shown in Fig. 1 is taken at $B_{\parallel} = 1$ mT. Up to around 7 mT, the shape of the arch does not change significantly, but above that value deviations appear, the flux arch gets broadened. This is not a problem for the responsivity, however, as we perform the measurements at similar responsivities for all in-plane fields by adjusting the flux bias value. The origin for this change with higher in-plane fields is not fully clear at this point, and we included the discussion of possible reasons in the SI.

To make more clear to the reader how the flux tuning depends on the in-plane field, however, we show and discuss corresponding data together with the data on field-dependence of the cavity linewidth (cf. previous point) in the new manuscript version SI.

In addition, we added a sentence to the main paper and caption about the arch exceeding one flux quantum, as requested by the reviewer. Also, we replaced the statement about constant flux bias by constant responsivity in the discussion of the in-plane dependence, as the responsivity is what we actually kept constant.

We note, however, that the “periodicity” of the flux dependence is still $1\Phi_0$ (cf. Fig. S6), each arch is just widened such that they overlap and multiple flux branches/states exist simultaneously for a single bias flux value.

Reviewer #1:

“3) Also related to point 2, the authors discuss various calibration techniques for Φ_b in the SI. As Φ_b seems to be generated by the on-chip flux bias line, I encourage the authors to numerically compute the magnetic field (e.g. using Biot-Savard) from the bias current. Does this way of calibration fits with the γ_L stated in the SI ? This represents a very critical point as the responsivity is directly connected to the magnetic field / flux calibration.”

Reply:

In principle, we agree with the reviewer on this point. And actually we did already previously what the reviewer suggests and computed the flux coupled into the SQUID from the bias line. In doing so, we found a significant mismatch. We also realized that we get a very good match with the observed periodicity (cf. Fig. S6), if we assume that the bias current flows directly through the SQUID. This is actually possible in our device, as both, the flux bias line as well as both inductors from the SQUID are connected to the chip ground plane.

To address this, we have modified the manuscript and now in the SI, we explicitly state the extracted current-to-flux conversion factor, and mention that it does not match with simple calculations of the flux bias line current. We suspect that the current flows directly through the SQUID as return currents to ground.

We also point out that these simulations and calibrations are not necessary for the determination of the responsivity as the flux axis is calibrated using the flux periodicity.

Reviewer #1:

“4) Page 2 paragraph starting with “The concept of coupling mechanical ...”. Theoretical proposals of squid induced optomechanical coupling date back to 2007 (Phys Rev B 76, 014511 (2007), Phys Rev B 78, 104516 (2008). Phys Rev A 93, 022510 (2016)) . Those works should also get credit.”

Reply:

Indeed and to our surprise, we overlooked the first two of these references when preparing the manuscript and are happy to include them in the new version of the manuscript. The third reference mentioned by the reviewer was already included.

Reviewer #1:

“5) Thermal noise spectrum given in Fig 2c (inset). Given the excitation scheme with the flux bias line, the authors should have a calibration technique to relate the magnetic flux fluctuations to the frequency fluctuations of the microwave resonator. To be specific, as the authors know the resonance frequency modulation for a given VNA “current amplitude”, they should be able to convert this amplitude to a flux amplitude (again using e.g. modeling of the stray-field of the line). This should give directly the calibration factor relevant for converting the flux fluctuations to a g_0 . What value of g_0 would the authors get for this spectrum and does it fit to the electromechanically induced transparency method?”

Reply:

When we record the thermal noise spectrum shown in Fig. 2c (inset), we do not modulate the SQUID flux. We only apply a single resonant microwave tone to the cavity and observe the sidebands generated by the mechanical fluctuations. Also, when we apply a flux modulation to the flux bias line, we at the same time excite the mechanical resonator to coherent oscillations as shown in Fig. 2c.

The main reasons, however, why we unfortunately cannot perform the suggested calibration is that we do not have a calibration of the MHz currents generated on the flux bias line. The DC current is sent by a DC current source and through DC lines to the chip and by that we know exactly the DC current flowing on the device. The MHz signal used to excite the mechanical resonance, however, is coupled to the device via a coaxial high-frequency input line with attenuators on each plate of the dilution refrigerator and combined with the DC signal in a bias-tee mounted to the mK plate. And as we do not have calibrated the attenuation on this high-frequency line for 7 MHz signals, we unfortunately do not know the MHz current amplitude on the chip.

Reviewer #1:

"6) In the context of point 5. Do the authors have any indications, that their mechanical resonator is in thermal equilibrium with the temperature of the fridge? Please comment."

Reply:

A rough estimation for the thermal occupation, added to the new version of the Supplementary Material, suggest that the mechanical mode temperature is about 50 mK, which implicates that it's not fully thermalized to the fridge base temperature.

Reviewer #1:

"7) When the authors discuss figure 2, they should state the flux-bias conditions responsivity and B_{parallel} . B_{parallel} of course only for panel c."

Reply:

We fully agree and we included it in the current version of the manuscript.

Reviewer #1:

"8) SI: Chapter S2.A: I noticed 2 circulators, a coupler and a filter in the line to the cryogenic hemt amplifier. In what sense are those included in the attenuation discussion? Typically, those can account (depending on model) for losses in the range of 2dB. How does that affect g_0 ?"

Reply:

We include in the original manuscript assumed 2 dB losses between the sample and the HEMT in our attenuation estimation in S2.A. This means that we assume the on-chip signal to be 2 dB larger than the signal arriving at the HEMT, which we in turn calibrate by the HEMT noise temperature and the detected signal-to noise ratio.

Also, we state that ultimately there will remain an estimated uncertainty in the photon number of about 3 dB, which can go both directions though. This means essentially a systematic uncertainty of ± 1.5 dB or roughly $\pm 0.2g_0$ for the extracted g_0 s.

We have modified this section to also discuss the implications of the possible systematic error on the estimates of g_0 .

Reviewer #1:

"9) Chapter S5 B: given the parameters stated in the manuscript for the $g_0/2\pi = 250\text{Hz}$ case, the "optical" damping described in (S49) should be on the order of the linewidth. Do the authors find the expected signatures of opto-mechanical "cooling"? This should also be directly visible in the transparency data. I suggest to extract Γ_{eff} and plot it vs the photon number to confirm the expectations."

Reply:

Unfortunately, we do not have systematic enough data to determine if our device shows signatures of optomechanical cooling.

The main reason for this is that in order to get a clean signal for optomechanically induced transparency, we have to operate the cavity usually with photon numbers close to the instability threshold on the unstable flux branch. For smaller photon numbers, the transparency window is very small and disappears in the signal noise and for larger photon numbers the cavity jumps to the stable flux branch. Most of our data are collected with similar cooperativities of 0.1 to 0.3. According to the theory, this only leads to optical damping rates 10-30% of the intrinsic mechanical linewidth.

While a change of linewidth of 30% should seem easy to determine, in these experiments, we observed that the mechanical linewidth seems not to be constant and seems to be fluctuating by up to a factor of 2 from experiment to experiment. Because the parameters (power, flux, cavity frequency, cavity linewidth) are strongly correlated, it becomes challenging to perform a systematic study determining the contribution of each of these parameters on mechanical linewidth changes. This is particularly difficult as for each measurement, the pulse tube is turned off, allowing 10 minutes of measurements before the fridge warmed up, which then subsequently required another 30 minutes to cool back down.

The limited range of powers before jumping out of the unstable flux branch limited our ability to perform systematic power dependence measurements, and the measurement of Γ_{eff} vs photon number was not possible.

We would like to note, however, that the method used for extracting the cooperativity, and estimating the g_0 , based on the circle diameters (see SI), does not require an independent measure of the intrinsic mechanical linewidth, and hence is not influenced by the fluctuations of the intrinsic linewidth.

Reviewer #1:

"10) Chapter S5 C: To the best of my knowledge, the Supplemental Information of the paper "Weiss et al., Science 330, 1520 (2010)" and the SI of "Zhou, X. et al., Nat Phys 9, 179 (2013)" contain a full description of the interference effect including both sidebands. As far as I understand, the authors in Weiss et al. are not using the resolved assumption in S23. Same is should be the case for S27 in Zhou. I wonder how the proposed fit methods in the manuscript by Rodrigues et al. relate to this transmission model. Please comment."

Reply:

Similar to the two references mentioned by the reviewer, we do not use the sideband resolved limit either in our manuscript (which would also not be justified considering our system parameters). All three are fully equivalent, the two equations mentioned by the reviewer are equal to Eq. (S45) in our manuscript. The only approximation we make to get the closed form expressions we use for the final analysis is that the mechanical linewidth is much smaller than the cavity linewidth, which is well justified as $\Gamma_m/\kappa \approx 10^{-6}$.

In both references, however, they assume the resolved-sideband limit for their further analysis and do not derive final closed form expressions for the transparency window in the general case.

We do so and instead of using just the amplitude of the transparency window, we use the circle diameter of the resonance in the complex plane. When the transparency window is exactly on

resonance of the cavity, the two methods are 100% equivalent. In the case of small detunings, however, the amplitude method is not straightforward anymore, because the anchor point of the OMIT circle is moved along the cavity resonance circle. We can easily take that into account by the factor given in Eq. (S61). For clarification, we also added this factor into Fig. S10f.

Reviewer #1:

"11) Chapter S5 D 1: Just out of curiosity, why do the authors stop the pulse tube ? They should spend a sentence what they prevent here experimentally."

Reply:

With the pulse tube running, we observe strong excess flux noise in the cavity and find that the cavity jumps more frequently to the stable flux branch. We suspect that the origin of this noise can be found in the details of how our system parts are mounted. Our high-field magnet is mounted to the 4 K plate, while the sample is mounted to the mixing chamber. Hence, mechanical vibrations can lead to relative motion between the magnet-coil and the sample, which possibly generates a considerable amount of time-varying flux in the SQUID by changing the parasitic out-of-plane component of our in-plane field. Just to visualize and quantify this problem a bit: If our chip vibrationally "rotates" by only 10^{-2} degrees with respect to the in-plane field axis in a field of 1 mT, it generates a flux modulation of about $10 \text{ m}\Phi_0$ in the SQUID.

We have added a clarifying comment to the corresponding section of the SI.

Reviewer #1:

"12) Fig S9. The authors should include the squid bias conditions, the photon number and the B_{parallel} for this dataset. Is the extracted effective linewidth consistent with the γ_{eff} expected by (S49)."

Reply:

Including the experimental parameters here is a very good suggestion and we have implemented with it in the modified version of the manuscript.

As in this dataset, the cooperativity is only about 0.1, the expected optical contribution to the total linewidth is $<10\%$ of the intrinsic linewidth and due to the aforementioned linewidth fluctuations it is not possible to make a sound statement about a possible agreement with the expectations from the theoretical expectations.

Reviewer #1:

"Technical points:

13) Fig S9: typo at "b Dta a and fit..." should read "Data..."

14) SI Page 8: second last line. There is a 2π missing in the equation of ω_2 "

Reply:

We thank the referee for spotting these typos, and have corrected them in the new manuscript version.

Reply to reviewer #2:

Reviewer #2:

"The experimental work is of high quality and is clearly presented both in the text and in a series of nice figures. The analysis of the data is done with care and appears valid. The resulting conclusions are well-supported and based on valid assumptions. A thorough supplementary section describes the experimental details. In summary, the paper provides an excellent proof of principle for a flux-mediated optomechanical device."

Reply:

We are grateful to the reviewer for their positive assessment and appreciation of our manuscript.

Reviewer #2:

"For all of these reasons, I find the paper appropriate for publication once the authors have addressed a few key questions, mostly related to the proof-of-principle nature of the result, i.e. the authors should better spell out what the ultimate limitations of this technique are and what exactly could be achieved with the ideal device."

"1) In the first paragraph of page 2, the authors state that unlike capacitive optomechanical cavities, which are limited by geometric and technological restrictions, their cavities may be able to reach the single-photon strong coupling regime. What are then the restrictions that limit their flux-mediated scheme? How far, in principle, could such a scheme push the coupling rates? What would have to be optimized?"

Reply:

To understand the limitations of the scheme, it is useful to look at equation 2 from the manuscript:

$$g_0 = B_{||} * d\omega/d\Phi * x_{zpf} * L$$

Here, there are four parameters to optimize.

To get as high a coupling as possible, a large $B_{||}$ should be used. The limitation in $B_{||}$ is determined by the maximum magnetic field that the superconducting circuit can sustain. The limitation of $B_{||}$ depends on the geometry of the film and junctions, on the material, and on how close to perfectly parallel the field can be applied, as typically out-of-plane fields are strongly detrimental to the circuit. For thin Aluminum, films of tens of nm can sustain fields of up to 100 mT. Thin films of materials such as NbTiN can sustain fields of up to 6T, even with low microwave loss (Samkharadze *et al.*, Physical Review Applied **5**, 044004 (2016)).

As discussed in the conclusion section of the manuscript extensively, another route to increasing g_0 is to increase $d\omega/d\Phi$. Our current design has a low $d\omega/d\Phi = 70 \text{ MHz}/\Phi_0$ due to the loop geometric and kinetic inductance and the series linear inductance. Optimizing the SQUID design, SQUID cavities with responsivities on the order of $10 \text{ GHz}/\Phi_0$ have been demonstrated.

Finally, the beam geometry can be changed to increase its length L and/or increase x_{zpf} . This will also increase the coupling, but one also has to be careful about the increase of the loop inductance, which can have a detrimental effect on $d\omega/d\Phi$.

As discussed in the manuscript, combining these considerations, we believe that there is significant room to increase g_0 by several orders of magnitude in future experiments.

A discussion of these points above are already present in the submitted manuscript in detail:

"The largest single-photon coupling rate we achieve here $g_0 \approx 2\pi \cdot 230 \text{ Hz}$ is comparable with the best values obtained for highly optimized capacitively coupled devices. As it is possible to achieve responsivities of several GHz/Φ_0 with SQUID cavities, we expect that with an optimized cavity it is possible to boost the single-photon coupling rates to the order of $\sim 10 \text{ kHz}$ per mT of in-plane field."

"Due to an imperfect alignment between the chip and the in-plane field, a considerable out-of-plane component was present and, most probably by introducing vortices, strongly influenced the properties of the cavities above $B_{||} = 10$ mT. Using a vector magnet to compensate for possible misalignments will allow to go up to about 100 mT with thin film Aluminum devices^{32,33} resulting in rate of $g_0 \approx$ MHz. When extending our materials to other superconductors such as Niobium or Niobium alloys, where similar constriction type SQUIDs have recently been used for tunable resonators³⁴, the possible field range for the in-plane field increases up to the Tesla regime."

"The single-photon coupling rates achieved with this first device are already competing with the best electromechanical systems and can be boosted towards the MHz regime by optimizing flux responsivity and applying higher magnetic in-plane fields. In addition, reducing the cavity linewidth to values of ~ 100 kHz will lead us directly into the single-photon strong-coupling regime [...]."

We felt that these already pretty clearly describe the limitations of the scheme, and could not find a clearer way of describing them. If the reviewer has specific considerations that should be addressed that are not already discussed, we would be happy to consider adding them to the manuscript and/or SI.

Reviewer #2:

"2) As stated in the 2nd to last paragraph of page 5, the authors largest achieved single-photon coupling rate is $2 \sqrt{\pi} 230$ Hz, which is comparable to the rates achieved by best conventional devices. They then claim that it should be possible to boost this rate by about 3 orders of magnitude, because SQUID cavities can be made with responsivities of several GHz/ Φ_0 . From Figure 1e, it looks like their cavity has about 10 MHz/ Φ_0 . Given that a 3 order of magnitude boost in coupling is a big claim, it should be further justified.

What steps would have to be taken to achieve such the required responsivity?"

To achieve a higher flux responsivity, the screening parameter $\beta_L = L_{loop} / (\pi * L_J) = 2 * L_{loop} * I_c / \Phi_0$, needs to be reduced. This can be done by either decreasing L_{loop} , by increasing L_J (decreasing I_c), or a combination of the two.

We added a clarifying sentence into the manuscript.

"What has to be improved relative to the presented device?"

Our current device has β_L of approximately 5. This should be reduced ideally to a value less than 0.1 to make screening effect negligible, which would give the require responsivity.

"Can this be realized in a device with the functionality of the one demonstrated in the manuscript?"

Yes, we believe so, by reducing I_c and reducing L_{loop} .

"What are the limitations?"

As mentioned above, the limitation in our current device is β_L . We outline strategies above to reduce β_L by reducing I_c and L_{loop} . Reducing I_c requires changes to the design of the nanobridge junctions with reduced I_c by using narrower or thinner junctions, and optimizing further to reduce the (geometric and kinetic) loop inductance. We believe that these can both be achieved with the current architecture, and the current architecture is not limited to reaching these responsivities.

We also note that SQUID cavities with more optimised β_L have been demonstrated to have flux responsivities of several (up to 10s of) GHz (cf. e.g. Yamamoto *et al.*, Appl. Phys. Lett. **93**, 042510 (2008) in SIS junctions, Levenson-Falk *et al.*, Appl. Phys. Lett. **98**, 123115 (2011) in nanobridge junctions).

Reviewer #2:

"3) In the conclusion, the authors restate the possibility of using their scheme to achieve single-photon strong coupling. However, the description of what this means and what could be achieved

with such devices is rather vague and devoid of references. The non-expert reader would benefit from a bit more clarification. What exactly are the "new type of devices and experiments" that can be realized? Why are microwave qubits with a nonlinearity induced by mechanics useful/interesting? What is the point of generating mechanical quantum states or photon blockade?"

Reply:

Yes, we agree that we could have clarified our statements and should have given the corresponding references at this point (once more after the introduction) and do so in the modified version of the manuscript with old and new references and additional statements.

Reviewers' Comments:

Reviewer #1:

Remarks to the Author:

Dear Editor,

the authors have addressed all my points and included a discussion about them in the manuscript / SI. The manuscript has improved and I therefore recommend publication in Nature Communications

Reviewer #2:

Remarks to the Author:

The authors have satisfactorily responded to the comments of the 2 referees and therefore the manuscript is now appropriate for publication.

Reply to the reviewers comments

Reply to reviewer #1:

"Dear Editor,

the authors have addressed all my points and included a discussion about them in the manuscript / SI. The manuscript has improved and I therefore recommend publication in Nature Communications"

Reply:

We thank the reviewer for the reading of the modified manuscript/SI and are very glad to know that we have addressed all questions and suggested modifications adequately. We would also like to thank the reviewer for their recommendation for publication in Nature Communications.

Reply to reviewer #2:

"The authors have satisfactorily responded to the comments of the 2 referees and therefore the manuscript is now appropriate for publication."

Reply:

We thank the reviewer for the positive feedback towards the manuscript/SI modifications and for their recommendation for publication in Nature Communications.